# Graphene-Based Localized Surface Plasmon Metasurface for Mid-Infrared Four-Band Ultra-Narrow Absorbing Sensor

**DOI:** 10.3390/s25247477

**Published:** 2025-12-09

**Authors:** Min Luo, Zihao Chen, Qiye Wen

**Affiliations:** 1School of Electronic Science and Engineering, University of Electronic Science and Technology of China, Chengdu 610054, China; 2Engineering Center of Integrated Optoelectronic & Radio Meta-Chips, University of Electronic Science and Technology, Chengdu 610054, China; 3Tianfu Jiangxi Laboratory, Chengdu 641419, China

**Keywords:** graphene, metamaterial, localized surface plasmon, mid-infrared, sensors

## Abstract

**Highlights:**

**What are the main findings?**

**What are the implications of the main findings?**

**Abstract:**

In this paper, the design of a mid-infrared four-band ultra-narrowband wave-absorbing sensor based on the localized equi-excited exciton resonance of graphene metamaterials is presented. The designed super-surface unit has a geometrically symmetric structure and is insensitive to incident light sources with different polarization directions. The absorbing sensor has four resonant wavelengths located at λ_1_ = 3.172 μm, λ_2_ = 3.525 μm, λ_3_ = 3.906 μm, and λ_4_ = 4.588 μm, with absorption efficiencies of 99.94%, 99.46%, 99.55%, and 98.16%, respectively. In addition, the dynamic tuning of the resonant wavelength and absorption efficiency can be realized by changing the gate voltage or through chemical doping of graphene. Moreover, the wave-absorbing performance can maintain stable absorption over a wide range of incidence angles from 0 to 50°. Finally, the wave-absorbing sensor was subjected to different ambient refractive indices, and the refractive index sensitivities corresponding to the four resonant wavelengths were obtained as 587.5 nm/RIU, 700.0 nm/RIU, 850.0 nm/RIU, and 900.0 nm/RIU, with FOM values of 48.96 RIU^−1^, 58.34 RIU^−1^, 53.13 RIU^−1^, and 28.13 RIU^−1^, respectively, all of which have superior sensing characteristics. Therefore, this paper enriches the variety of mid-infrared absorber sensors and has a broad application prospect in the fields of wave absorption, sensing, and detection.

## 1. Introduction

The mid-infrared band (3–5 μm) is a low-loss, weakly turbulent, and weakly background noise window of the atmosphere which can well overcome the influence of atmospheric molecules and is an ideal band for molecular sensing detection in space [1,2]. The mid-infrared window can be used not only for the detection of molecular content and the identification of molecular types, but also for the imaging of molecules, etc., thus having a wide range of applications in military, environmental monitoring, medical therapy, and basic research [3,4]. Since the infrared spectra of any two different biochemicals can hardly have the same absorption frequency, infrared spectroscopy can provide molecular “fingerprints” for detection and identification. In a variety of sensing applications, sensor technology based on absorbing wave forms can provide a powerful guarantee for nondestructive testing by utilizing the optical refractive index of biochemicals, which distinguishes them from other substances, enabling us to determine the compositional characteristics of the object to be detected [5,6]. In order to continuously improve the absorption efficiency and sensitivity of wave-absorbing sensors, researchers have developed various types of refractive index sensors by employing a variety of physical mechanisms, including Fano resonance, continuous-domain bound states, and surface plasmon resonance (SPR for short). For example, Liu et al. [7] designed a plasma refractive index sensor based on Fano resonance, with sharp Fano peaks due to symmetry breaking and coherent coupling between the bright and dark modes of the structure, which can achieve absorption efficiency up to a maximum sensitivity of 600 nm/RIU. Xiao et al. [8] investigated an inverted U-shaped resonance cavity with a Fano resonance-based sensor. In the same year, Jiang et al. [9] proposed a high-sensitivity ultra-surface iso-excitation structure with a maximum refractive index sensitivity of 598.9 nm/RIU in the near-infrared wavelength band; the structure based on the resonance absorption sensing of surface iso-excitation is highly sensitive in the near-field range due to its weak radiation characteristics, which makes the electromagnetic energy highly localized in the near-field range, with strong absorption properties and high sensitivity.

SPR is an electromagnetic mode generated by the interaction between light fields and free electrons in metals [10], and this interaction allows the light fields to be trapped by the collective oscillations of the free electrons, forming a special local surface plasmon resonance (LSPR for short.) The localized electromagnetic field properties of LSPR allow it to break through the limit of conventional optical diffraction and to facilitate the localized enhancement of the electromagnetic field [11]. In this regard, graphene and noble metal materials have similarities in the mid-infrared and terahertz bands, and their surfaces can also support SPP propagation [12]. Compared with metal surface plasma oscillations, graphene metamaterial-based localized equipartitioned exciton resonance has more advantages [13,14], such as follows: ultra-narrowband absorption can be achieved thanks to great mode confinement, and the propagation distance is long in the infrared region. Meanwhile, graphene’s conductivity properties can be tuned by adjusting the Fermi energy level and bias electric field, among other methods [15,16]. Moreover, with the development of nondestructive testing technology, the demand for high-performance wave-absorbing sensors increases, and traditional sensors face difficulty in meeting the demand [17]. Therefore, ultra-narrowband wave-absorbing sensors have become the focus of researchers’ attention, and at the same time, the application fields of wave-absorbing sensors are being broadened [18]. However, most of the current wave-absorbing sensors are mainly single-band and dual-band absorbing resonance points, while relatively few studies have been carried out on multiband ultra-narrowband perfect sensors. This is mainly due to the strict equivalent impedance matching condition limitation, which makes it difficult to design multiband narrowband perfect sensors with simple hypersurface structures [19]. In addition, it is difficult to fabricate wave-absorbing sensors with complex structures, as they cannot be dynamically tuned [20].

Therefore, to overcome the shortcomings of the existing technology, this paper proposes a mid-infrared four-band ultra-narrowband wave-absorbing sensor based on the LSPR of graphene metamaterials. A simple graphene hypersurface sensor structure is obtained by utilizing the localized equipartitioned exciton properties of patterned graphene metamaterials. It is characterized by multiband, tunable harmonics, high sensitivity, and a high quality factor with maximum sensitivity of 900 nm/RIU and 58.33 RIU^−1^, respectively. The sensor also has polarization-independent and stable absorption characteristics even with light sources of large angles of oblique incidence.

## 2. Model Structure and Theory

Presented herein is a graphene-based localized surface plasmon metasurface for a mid-infrared four-band ultra-narrowband wave-absorbing sensor, which consists of a plurality of ultra-surface units. The wave-absorbing sensor units include a bottom metal film, a dielectric layer, and a graphene layer, in ascending order; adjacent layers are adhered to each other and their geometric centers are all in a straight line, as shown in Figure 1a. In this case, the cross section of each wave-absorbing sensor cell is square and the period of the basic cell *P_x_* = *P_y_* = 600 nm. The underlying metal can be gold, silver, copper, or aluminum, and its dielectric constant can be expressed by the Drude model as follows [21,22]:(1)$$\sigma (\omega )=\varepsilon_{\text{∞}}-\frac{{\omega_{p}}^{2}}{\omega^{2}+j\omega \gamma }$$
where $\varepsilon_{\text{∞}}=9.1$ is the dielectric constant at infinite frequency, $\omega_{p}=1.33\;{\text{×}\;10}^{14}\;\mathrm{rad}/s$ is the frequency of equipartitioned exciton oscillation, $\gamma =1.47\;{\text{×}\;10}^{16}\;\mathrm{rad}/s$ is the damping rate associated with absorption attenuation, and the thickness of the underlying metal layer H_2_ is 200 nm. The intermediate dielectric layer is silica (SiO_2_) with an effective dielectric constant ε = 3.53 and a thickness H_1_ of 1470 nm. The top graphene layer has many advantages as a metamaterial, and the intrinsic graphene has zero band gap, which makes the intrinsic graphene exhibit metallic properties. Further, the band gap and Fermi energy levels of graphene can be effectively tuned by applying an external gate voltage or chemically doping the graphene. Changing its conductivity enables the construction of dynamically tuned devices. The total conductivity of graphene can be expressed as $\sigma_{g}=\sigma_{\mathrm{intra}}$ + $\sigma_{\mathrm{inter}}$. $\sigma_{\mathrm{intra}}$ and $\sigma_{\mathrm{inter}}$ represent the intraband and interband conductance, respectively. According to Kube’s formula, the intraband and interband conductivity of graphene can be expressed in the following forms [23,24].(2)$$\sigma_{\mathrm{intra}}=\frac{ie^{2}K_{B}T}{\pi \hbar^{2}\left(\omega +i\tau^{-1}\right)}\left\{\frac{E_{f}}{K_{B}T}+2\ln\left[\exp(-\frac{E_{f}}{K_{B}T})+1\right]\right\}$$(3)$$\sigma_{\mathrm{inter}}=\frac{ie^{2}}{4\pi \hbar^{2}}\ln\left[\frac{2|E_{f}|-\hbar \left(\omega +i\tau^{-1}\right)}{2|E_{f}|+\hbar \left(\omega +i\tau^{-1}\right)}\right]$$
where *e* = 1.6 × 10^−19^ C refers to the charge of the electron, *K*_B_ refers to the Boltzmann constant, and *ħ* = *h/*2*π* = 1.05 × 10^−34^ J·s, which is the approximate Planck’s constant. *T* refers to the ambient temperature, while ω refers to the angular frequency of the incident wave. $E_{f}$ = 0.74–0.86 eV and τ = 0.5~2.0 Ps refer to the Fermi energy level and relaxation time of the graphene layer, respectively. Under the conditions of the electromagnetic band (3–5 μm, corresponding to *ħ*ω ≈ 0.25–0.41 eV) and the $E_{f}$ = 0.74–0.86 eV used in this study, $2|E_{f}|$ > ℏω, which satisfies the Pauli blocking condition, and the interband conductance is suppressed [25]. Therefore, the total conductance of the graphene metamaterial can be approximately expressed as follows [26]:(4)$$\sigma_{g}\left(\omega \right)=\frac{ie^{2}|E_{f}|}{\pi \hbar^{2}\left(\omega +i\tau^{-1}\right)}$$

Next, the dielectric constant of graphene metamaterials can be expressed by the following equation [27,28]:(5)$$\varepsilon (w)=1+i\sigma_{g}\left(\omega \right)/\omega \varepsilon_{0}t$$
where ε_0_ represents the vacuum dielectric constant, and t represents the thickness of the graphene layer. When the thickness of a single layer of graphene is t = 0.334 nm, the imaginary part of the dielectric fully converges in the limit of t→0, and the surface conductance $\sigma_{g}\left(\omega \right)$ itself determines the electromagnetic response [29]. Therefore, in this paper, the thickness of a single layer of graphene is set to 0.334 nm for the equivalent mapping of $\sigma_{g}\left(\omega \right)$→ε(w) in the simulation calculation, and this setting will not affect the electromagnetic response results. According to Equations (4) and (5), it can be seen that the electrical conductivity σ(ω) of graphene can be adjusted by changing the Fermi level and relaxation time without changing its geometry. The Fermi level is dynamically adjusted by applying an external gate voltage, while the relaxation time of graphene depends mainly on the manufacturing process and can be changed via chemical doping during preparation. Therefore, the continuous broadband tunable electromagnetic property of graphene metamaterials simplifies the design of optoelectronic devices and increases the flexibility of their use. In this paper, the electromagnetic frequency characteristics of graphene are programmed using Matlab 2020b software. Subsequently, they are imported into the Lumerical Finite Difference Time Domain (FDTD) solution 2024 R2 software in the form of two-dimensional surface conductivity (frequency-dependent conductivity), where the electrical behavior of graphene is completely determined by σ(ω). Figure 1b presents a top-view schematic of the four-band ultra-narrowband wave-absorbing sensor. In this study, the shape and size of the structure were achieved by parametrically controlling the scanning cell period, dielectric layer thickness, and the radius and length/width of the ring/box structure. This design enables the structure to achieve impedance matching at four operating wavelengths, resulting in near-perfect absorption, high sensitivity, and multimode stability in mid-infrared sensing performance. Finally, the graphene metamaterial is patterned by air etching, and the two air rings are etched with widths of W_1_ = 75 nm and W_2_ = 40 nm, outer diameters of R_1_ = 125 nm and R_2_ = 240 nm, and surrounding squares with lengths and widths of L_1_ = 120 nm and L_2_ = 170 nm, respectively. In the fabrication of actual devices, a SiO_2_ dielectric layer can first be deposited on a metal substrate using physical vapor deposition. Subsequently, a monolayer of graphene is grown using chemical vapor deposition and transferred to the SiO_2_ surface using a wet transfer process. Finally, a monolayer patterned graphene structure is obtained using electron beam lithography and oxygen plasma etching [30]. Figure 1c presents a top-view schematic of the localized isotropic three-dimensional array structure of the excitation element sensor. During the simulation, periodic boundary conditions are selected for the sensor unit in the X and Y directions, and 24 perfectly matched layers (PMLs) are added in the Z-axis direction. The simulation time is set to 8000 fs with an adaptive time step. The mesh size in the X and Y directions is 0.005 μm, and the mesh size in the Z direction is 0.0025 μm. The incident light wavelength range is 3.0 to 5.0 μm, and the incident light direction is perpendicular to the XY directions, indicating downward incidence. Furthermore, the distance between the monitoring port and the sensor unit is 4 μm, which is greater than the minimum wavelength.

Further, the absorption efficiency of the wave-absorbing sensor in this paper can be calculated by the following equation [21,30]:(6)$$A(\lambda )=1-T(\lambda )-R(\lambda )=1-{\left|S_{21}\left(\lambda \right)\right|}^{2}-{\left|S_{11}\left(\lambda \right)\right|}^{2}$$
where $S_{21}\left(\lambda \right)$ and $S_{11}\left(\lambda \right)$ are the transmission amplitude and reflection amplitude, respectively. T(λ) and R(λ) are the transmittance and reflectance, respectively. Obviously, the absorption efficiency A(λ) is maximized when T(λ) and R(λ) are sufficiently small. Since the thickness of the metal reflective layer in the present invention is much larger than its skin depth, which is sufficient to inhibit the transmission of all waves, T(λ) is almost zero. Therefore, A(λ) in the present invention can be derived from the following equation [31]:(7)A(λ)=1−R(λ)

That is, perfect absorption is achieved when R(λ) is close to zero.

On this basis, the absorbing sensor based on graphene metamaterials with localized equipartitioned exciton resonance described in this paper achieves four-wavelength ultra-narrowband perfect resonance absorption peaks in the mid-infrared region, as shown in Figure 1d. The corresponding four resonance wavelengths are at λ_1_ = 3.172 μm, λ_2_ = 3.525 μm, λ_3_ = 3.906 μm, and λ_4_ = 4.588 μm, with absorption efficiencies of 99.84%, 99.46%, 99.55%, and 98.16%, respectively. Based on the four-band perfect absorption of the absorbing sensor device, the position of different resonance bands can be detected in order to achieve contactless sensing with a high quality factor and high sensitivity.

## 3. Results and Analysis

In order to investigate the physical mechanisms of the absorption peaks in the above four bands, the electric field distributions on the graphene hypersurface were calculated at λ_1_ = 3.172 μm, λ_2_ = 3.525 μm, λ_3_ = 3.906 μm, and λ_4_ = 4.588 μm, respectively, as shown in Figure 2. When the incident light wavelength is λ_1_ = 3.172 μm, the electric field intensity is mainly confined to the inner ring in the center of the graphene layer, and due to the coupling of the patterned graphene to the electric field and the provision of the electric dipole resonance, which oscillates in anti-phase with the metal film to form the localized plasmonic resonance absorption, the distribution of the electric field is as shown in Figure 2a. When the wavelength of incident light is λ_2_ = 3.525 μm and λ_3_ = 3.906 μm, the electric field is mainly localized in the air-etched outer ring and square region, and the air holes etched in the middle enhance the plasma resonance on the patterned graphene surface, which in turn distributes the electric field on its surface at the periphery of the super-surface, resulting in electric dipole resonance of the incident electromagnetic wave around the graphene monolayer, with the distribution of the electric field shown in Figure 2b,c. When the wavelength of the incident light is λ_4_ = 4.588 μm, the plasma resonance of the graphene super-surface is enhanced, which leads to the distribution of the surface electric field in the inner part of the air-etched outer ring, resulting in the electric dipole resonance of the incident electromagnetic wave around the graphene layer, which is then absorbed by the local electromagnetic field of the graphene surface, and the distribution of the electric field is as shown in Figure 2d. When the incident electromagnetic wave energy is localized in the hypersurface structure, the electromagnetic wave transmitted from the graphene surface will be reflected in the SiO_2_ layer, and the superimposed multiple reflections can increase the absorption of electromagnetic wave energy in the structure; thus, perfect absorption can be realized. At this point, the reflectivity R(λ) is almost 0. Combined with Equation (7), it can be concluded that the electromagnetic wave is absorbed by the local electromagnetic field on the graphene surface [32]. It can be seen that four resonance wavelengths can be formed by the air circle and square shapes, and the reflectivity is greatly reduced, thus realizing a four-band absorption peak.

Impedance matching is an important factor in ensuring the perfect coupling between the absorber and the incident electromagnetic waves [33]. According to the impedance matching theory of metamaterial absorbing sensors, the following equation is defined [34,35]:(8)$$Z=\frac{Z_{\mathrm{in}}}{Z_{0}}=\sqrt{\frac{{(1+S_{11})}^{2}-S_{21}^{2}}{{(1-S_{11})}^{2}-S_{21}^{2}}}$$

Therefore, we can obtain the equivalent input impedance Z of this absorber from the simulation results, as shown in Figure 3. When the patterned graphene layer enables the input impedance of the absorber Z_in_ to match the free-space impedance Z_0_, the system’s effective impedance satisfies Re(Z) ≈ 1 and Im(Z) ≈ 0 [36,37]. Under this condition, the reflection coefficient is strongly suppressed (S_11_ ≈ 0), leading to near-perfect absorption. By examining the input–impedance matching behavior together with the simulated absorption spectra, it is evident that the proposed sensing absorber achieves excellent impedance matching at the resonant wavelengths λ_1_ = 3.172 μm, λ_2_ = 3.525 μm, λ_3_ = 3.906 μm, and λ_4_ = 4.588 μm. Moreover, when the real and imaginary parts of the equivalent impedance Z deviate from 1 and 0, respectively, the absorption efficiency decreases sharply, thus realizing four-band perfect absorption in the ultra-narrowband sensor.

For wave-absorbing sensors with fixed structural parameters, dynamic tunability has more important application value. In particular, existing research has shown that the Fermi level and relaxation time of graphene can be altered within a certain range through gate voltage modulation or chemical doping [38,39]. The formula for tuning the $E_{f}$ of graphene with an applied voltage is as follows [40,41].(9)$$E_{f}=V_{f}\sqrt{\pi \varepsilon_{0}\varepsilon_{r}V_{g}/et_{s}}$$
where V_g_ is the applied voltage (which can be modulated by changing the gate voltage or through chemical doping), V_f_ is the Fermi velocity (V_f_ = *c*/300, with *c* being the speed of light in vacuum), t_s_ is the thickness of the dielectric layer, and $\varepsilon_{r}$ denotes the relative permittivity. Patterned graphene serves as the top gate electrode, while the metal reflective layer can simultaneously act as the bottom electrode, electrically controlling the gate voltage. In this study, the Fermi level E_f_ of the graphene is tuned in the range of 0.74–0.86 eV. Substituting this into Equation (9) yields the required gate voltage V_g_, which is approximately 18–29 V. Figure 4a shows that as the Fermi level increases, the resonant wavelengths all exhibit a blue shift, with modulation ranges including 3.079–3.273 μm, 3.420–3.640 μm, 3.791–4.032 μm, and 4.459–4.733 μm. Meanwhile, the modulation depths are, respectively, 0.194 μm, 0.220 μm, 0.241 μm, and 0.274 μm. Further, according to Equation (10), the electronic relaxation time *τ* can also be modulated to dynamically regulate the conductivity of graphene [42]:(10)$$\tau =E_{f}v/\left(ev_{f}^{2}\right)$$
where *v* represents the carrier mobility of graphene. In the actual process, organic molecules can be added or removed on the surface of the graphene layer [43], which will significantly change the carrier mobility *v*, thus changing the relaxation time *τ*. Figure 4b presents the variation curve of the absorption spectrum of the absorbing sensor as the relaxation time *τ* increases from 0.5 Ps to 2.0 Ps. It can be seen that with the increase in the relaxation time, the absorption efficiencies at the four resonance points are effectively modulated and the resonance wavelength remains unchanged. The modulation ranges of absorption efficiency are 36.83~99.84%, 39.45~99.46%, 68.13~99.55%, and 83.11~98.16%, and the modulation depths are 63.01%, 60.01%, 31.42%, and 115.05%, respectively. Therefore, compared with precious metal hypersurface wave-absorbing sensors, graphene-based LSPR wave-absorbing sensor devices can effectively modulate the absorption spectra by modulating the Fermi energy level and relaxation time of the graphene layer without changing its geometry, making them more practical in the application process.

In practice, there is often not just a single vertically incident plane wave. Therefore, it is extremely important that the wave-absorbing sensor exhibits insensitivity to oblique incidence of the light source [34,44]. The absorption intensity change spectra of this sensor under TE polarization and TM polarization are investigated by varying the polarization angle of the incident light source when the angle of incidence increases from 0° to 50°, as shown in Figure 5. Obviously, the absorption spectra under TE and TM polarization are the same when the incident angles are both 0°, which is due to the fact that the localized surface plasmon resonance achieved based on the patterned structure of graphene metamaterials has a geometrically symmetrical structure in the XY plane, i.e., the two-dimensional plane in which the graphene layer is located, and it is insensitive to incident light sources with different polarization directions. Moreover, in the angular range of 0~50° of the incident light source, the absorber produces a slight blueshift phenomenon under TE polarization, but the absorption intensity remains almost unchanged. Meanwhile, under TM polarization, the absorption peaks at the two long wavelengths of the absorber gradually decrease at incident angles greater than 40°, but the absorption intensity stabilizes above 85% at 50°. This indicates that the ultra-narrowband absorbing sensor is insensitive to the incident angle in the range of 0° to 50°. The absorption spectra in the two polarization modes complete the ultra-narrowband perfect absorption at the resonance band, which meets the requirements of practical applications.

Based on the strong energy localization effect of LSPR, the device is highly sensitive to changes in the refractive index of the surrounding medium, and thus its resonance peak position will shift with the change in refractive index [45]. However, traditional single-band metamaterial sensors can only achieve single-point matching between the characteristic frequency of the analyte and the resonant frequency of the device, and the available spectral information is limited, resulting in limited detection accuracy and sensitivity [46]. In contrast, multiband metamaterial sensors can form multi-point matching with the “fingerprint” frequency band of the analyte at multiple resonance points and support multi-parameter collaborative measurement, which significantly improves identification accuracy and sensitivity. At the same time, this study introduces the dynamic adjustment of the Fermi level to achieve control of the resonant frequency band and intensity while keeping the structure unchanged, which combines multi-frequency resonance and tunability. Taking advantage of these properties, the four-band ultra-narrowband wave-absorbing sensor designed in this paper can be used as a refractive index sensor to detect the change in the refractive index of an object, enabling us to analyze the chemical composition of the measured object. To study the characteristics of the response of absorption sensors to changes in the external refractive index, this study gradually adjusts the environmental refractive index *n* in the range of 1.00–1.08 (with a step size of 0.02). This refractive index range is often used to characterize the general response of devices to refractive index perturbations [47]. The absorption spectra corresponding to five resonance modes with different external refractive indexes are obtained sequentially, and the four resonance bands of these five spectral intervals are named as Mode A, Mode B, Mode C, and Mode D, from left to right, as shown in Figure 6a. With the increase in the refractive index, the resonance absorption peaks of the four modes are red-shifted, and the half-peak full widths, FWHWs, are calculated to be 15.2 nm, 12.3 nm, 18.4 nm, and 32.1 nm for Mode A, Mode B, Mode C, and Mode D, respectively. The sensitivity of the sensor, S, and the quality factor, FOM, which are important parameters characterizing sensing performance, can be expressed in the following form [48,49]:(11)S=∆λ/∆n(12)FOM=S/FWHM
where Δλ denotes the amount of change in the absorption resonance wavelength of the sensor, Δn denotes the change in the refractive index of the external environment, and FWHM is the full width of the half peak at the absorption peak. Figure 6b shows a plot of Δn with the change in the refractive index of the external environment versus the change in Δλ at the wavelength position of the resonance peak. According to Equation (11), the slope of its fitted straight line represents the sensitivity of the sensor. The sensitivities of Mode A, Mode B, Mode C, and Mode D are calculated to be 587.5 nm/RIU, 700.0 nm/RIU, 850.0 nm/RIU, and 900.0 nm/RIU, respectively. In addition, the absorption peaks of Mode A, Mode B, and Mode C decrease with the increase in the ambient refractive index, but all of them are maintained at near-perfect absorption of more than 95%, and the peak absorption of Mode D gradually increases. The quality factor quantifies the response characteristics of the sensor, and it is affected by the thickness of the medium layer of the absorbing sensing, which is related to the sensitivity and half-peak full width at different resonance wavelengths [50]. As shown in Figure 7a–d, the maximum quality factors of the four modal resonance peaks are calculated to be 48.96 RIU^−1^, 58.34 RIU^−1^, 53.13 RIU^−1^, and 28.13 RIU^−1^, according to Equation (12), where RIU is the refractive index unit (RIU). Compared with the previously studied sensors, the wave-absorbing sensor described in this paper has superior absorption efficiency, sensitivity, and FOM, and is dynamically tunable, polarization-independent, and angle-insensitive, as shown in Table 1 [51,52,53,54,55]. The results indicate that the wave-absorbing sensor has a broader implementation prospect in practical applications.

## 4. Conclusions

In conclusion, the design of a mid-infrared four-band ultra-narrowband absorption sensor based on graphene metamaterials is introduced. The sensor structure consists of a bottom metal film, an intermediate dielectric layer, and a top graphene layer, and its geometric configuration is symmetrical, thus ensuring high insensitivity to incident light with different polarization directions. At four specific resonance wavelengths λ_1_ = 3.172 μm, λ_2_ = 3.525 μm, λ_3_ = 3.906 μm, and λ_4_ = 4.588 μm, the absorption efficiencies are as high as 99.94%, 99.46%, 99.55%, and 98.16%, respectively. The research shows that the dynamic regulation of resonance wavelength and absorption efficiency can be achieved by adjusting the gate voltage of graphene or by employing chemical doping. In addition, the sensor can still maintain stable absorption characteristics in a wide range of incident angles from 0 to 50, which shows that it has good angular stability. With the change in the refractive index of the test environment, the maximum refractive index sensitivity of the sensor is 900.0 nm/RIU, and the maximum FOM is 58.34 RIU^−1^. Compared with similar sensors, the sensor has better sensing performance. To sum up, this study not only expands the design concept of mid-infrared absorption sensors, but also successfully achieves multiband operation, tunability, high sensitivity, and an excellent quality factor. Therefore, this design is expected to achieve breakthroughs in the fields of high-sensitivity gas detection and high-precision biomolecule recognition by virtue of the unique exciton resonance characteristics of graphene metamaterials, which will bring new possibilities for sensing applications in the mid-infrared band.

## Figures and Tables

**Figure 1 sensors-25-07477-f001:**
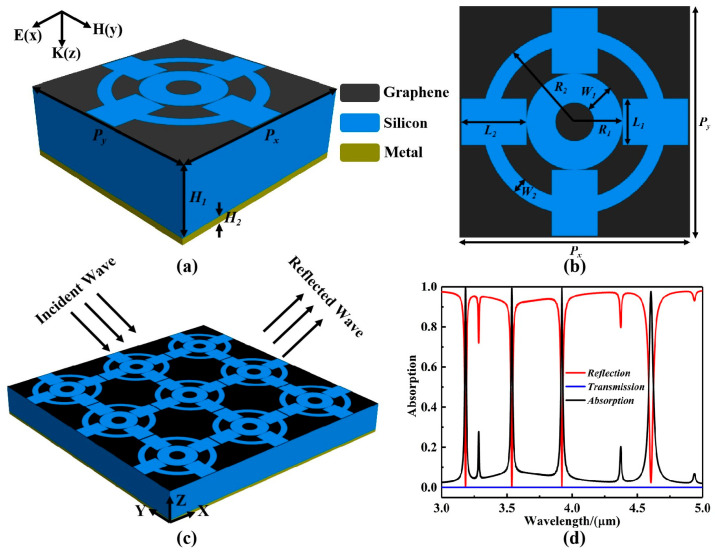
(**a**) The three-dimensional unit structure of the graphene-based localized surface plasmon metasurface for the mid-infrared four-band ultra-narrowband wave-absorbing sensor. (**b**) A top-view schematic diagram of the four-band ultra-narrowband wave sensor. (**c**) A schematic diagram of the periodic array structure of the four-band ultra-narrowband absorptive sensor. (**d**) The spectral radiative efficiency of the four-band ultra-narrowband absorptive sensor in the band of 3.0–5.0 μm, where the red solid line indicates reflection, the blue solid line indicates transmission, and the black solid line indicates absorption.

**Figure 2 sensors-25-07477-f002:**
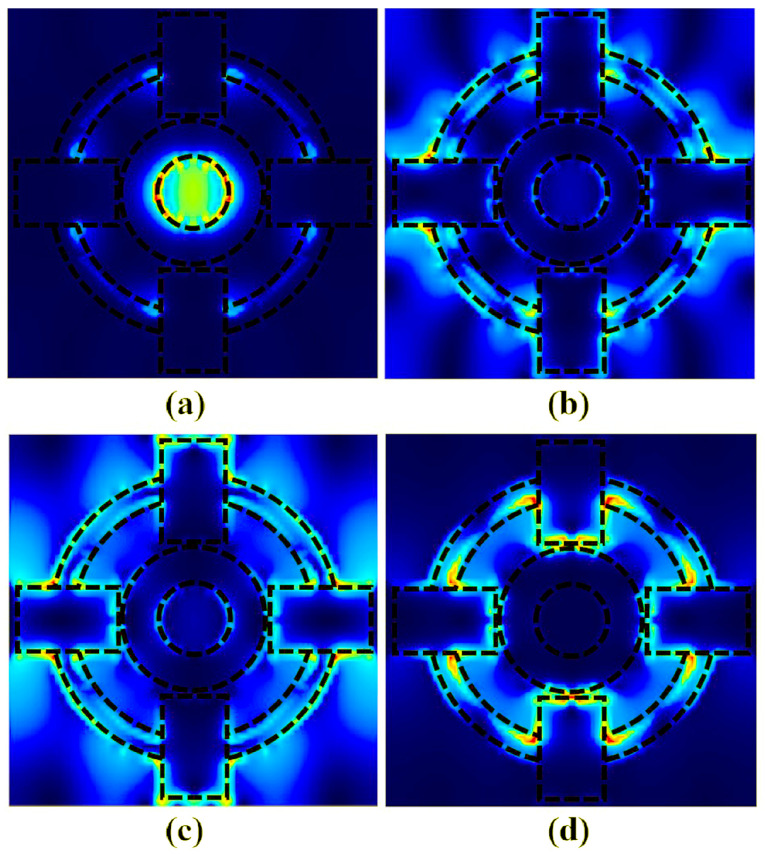
The cross-sectional electric field distributions of the four-band ultra-narrowband absorbing sensor in the XY plane of graphene at different resonance wavelengths (**a**) λ_1_ = 3.172 μm, (**b**) λ_2_ = 3.525 μm, (**c**) λ_3_ = 3.906 μm, and (**d**) λ_4_ = 4.588 μm.

**Figure 3 sensors-25-07477-f003:**
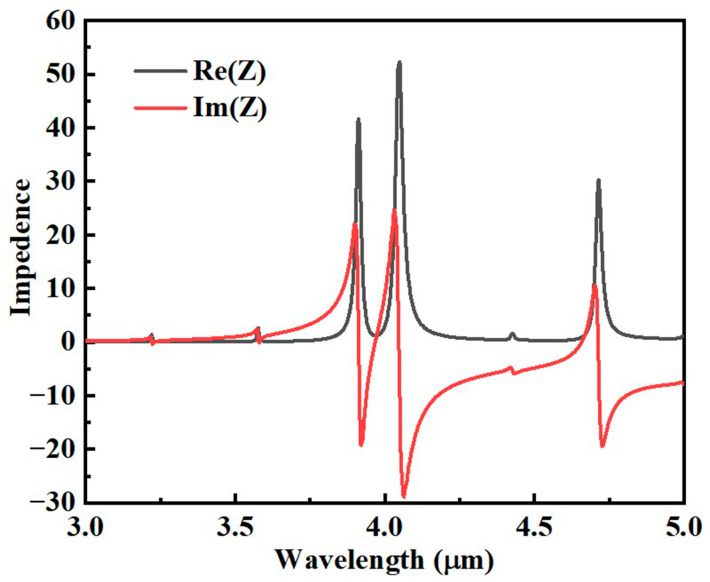
The real part Re(Z) and imaginary part Im(Z) of the effective impedance Z of the resulting wave-absorbing sensor.

**Figure 4 sensors-25-07477-f004:**
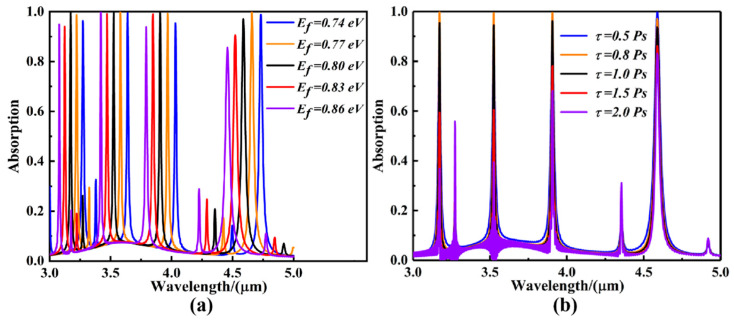
(**a**) Variation in the absorption curves of four-band ultra-narrowband absorption sensors during the process of increasing the Fermi energy level E_f_ of graphene from 0.74 eV to 0.86 eV. (**b**) Variation in the absorption curves of four-band ultra-narrowband absorption sensors during the process of increasing the relaxation time τ of graphene from 0.5 Ps to 2.0 Ps.

**Figure 5 sensors-25-07477-f005:**
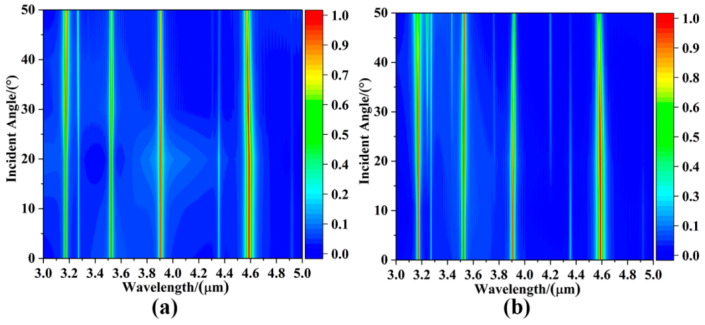
Absorption spectra of graphene-based localized surface plasmon metasurface for mid-infrared four-band ultra-narrow absorbing sensor under (**a**) TE polarization and (**b**) TM polarization, when the incident angle of the light source is increased from 0° to 50°.

**Figure 6 sensors-25-07477-f006:**
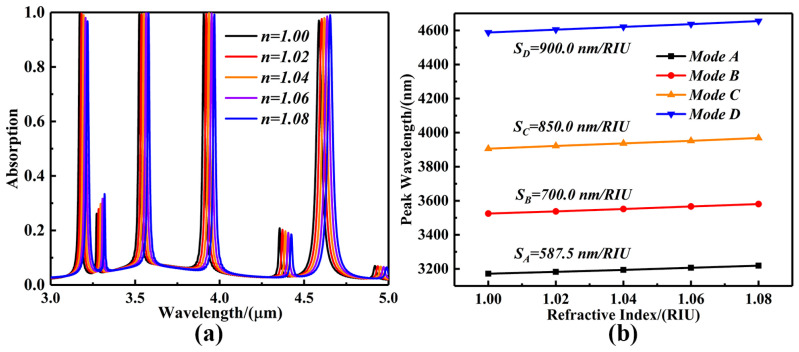
(**a**) A plot of the variation in the absorption spectra of the four-band ultra-narrowband wave-absorbing sensor as the ambient refractive index *n* varies from 1.00 to 1.08. (**b**) A schematic diagram of the linear fitting relationship between the resonance peak wavelength of the wave-absorbing sensor and the ambient refractive index.

**Figure 7 sensors-25-07477-f007:**
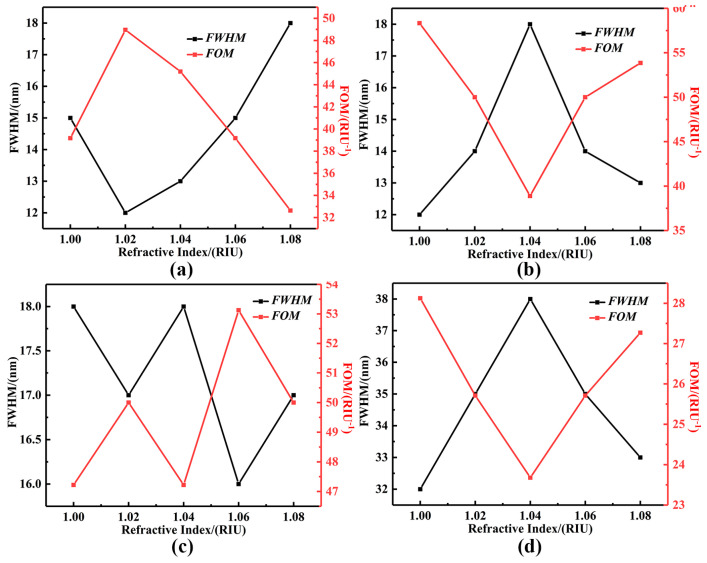
The curves of half-peak full width (FWHM) and the quality factor (FOM) of the mid-infrared four-waveband ultra-narrowband wave-absorbing sensor based on graphene metamaterials with localized equi-isotropic exciton resonance in (**a**) resonance mode A, (**b**) resonance mode B, (**c**) resonance mode C, and (**d**) resonance mode D.

**Table 1 sensors-25-07477-t001:** The sensor presented in this study compared with the sensors studied by predecessors.

References	[51]	[52]	[53]	[54]	[55]	Proposed
Year	2017	2019	2022	2022	2024	2025
Absorption	/	99.9%	99.9%	99.91%	99%	99.84%
Tunable	NO	NO	NO	YES	NO	YES
S (nm/RIU)	750	701	579	839.39	817	900
FOM (RIU^−1^)	65.2	/	12.46	54.03	12	58.34

## Data Availability

The dataset is available on request from the authors.

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
