# Peer review of "Graphene-Based Localized Surface Plasmon Metasurface for Mid-Infrared Four-Band Ultra-Narrow Absorbing Sensor"

_sensors, 2025, doi:10.3390/s25247477_

Round 1

Reviewer 1 Report

Comments and Suggestions for Authors

In the methodology section, a device model has been proposed. The model consists of a sandwich structure with a metal electrode at the bottom, a silicon layer in the middle, and a graphene layer on top. However, the specific thickness of the graphene layer used in this study is not mentioned. Investigating how variations in graphene thickness could influence sensitivity at different wavelengths would strengthen the overall quality of the manuscript.

Additionally, the type of silicon layer used—whether intrinsic, p-type, or n-type—has not been specified. Discussing how the selected silicon type might affect the infrared sensor’s detection capability and optical properties would provide valuable insight. Moreover, the criteria and conditions under which the infrared sensor structure was designed are not clearly described. Therefore, a more detailed discussion on the influence and selection of the geometric design criteria of the graphene layer is required.

And please describe in more detail the differences between the results obtained in this manuscript and REF 49, apart from the geometric structure of graphene.

Comments on the Quality of English Language

There are some syntax error and some assertive words

Author Response

Reviewer #1:

General Comments: In the methodology section, a device model has been proposed. The model consists of a sandwich structure with a metal electrode at the bottom, a silicon layer in the middle, and a graphene layer on top.

Response: Thanks for the reviewer’s positive comments, and we will answer the reviewer’s questions in detail below.

Comment 1: However, the specific thickness of the graphene layer used in this study is not mentioned. Investigating how variations in graphene thickness could influence sensitivity at different wavelengths would strengthen the overall quality of the manuscript.

Response: Thank you very much for your suggestions on our manuscript. In this study, monolayer graphene was used in modeling and simulation, with an equivalent thickness of t=0.334 nm, a value commonly used in the literature. The graphene thickness of t=0.334 nm and the reason for its use are clarified on Page 4, Lines 143-149 of the Revised Manuscript.

Where ε0 represents the vacuum dielectric constant, and  represents the thickness of the graphene layer. When the thickness of a single-layer graphene is t=0.334 nm, the imaginary part of the dielectric fully converges in the limit of t→0, and the surface conductance  itself truly determines the electromagnetic response [29]. Therefore, in this paper, the thickness of the single-layer graphene is set to 0.334 nm for the equivalent mapping of  in the simulation calculation, and this setting will not affect the electromagnetic response results.

29.M. Yi, J. Wang, A. Li, Y. Xin, Y. Pang, Y. Zou, Conductive micropatterns containing photoinduced in situ reduced graphene oxide prepared by ultraviolet photolithography. Adv. Mater. Technol., 8 (2023) 2201939.  https://doi.org/10.1002/admt.202201939

Comment 2: Additionally, the type of silicon layer used—whether intrinsic, p-type, or n-type—has not been specified. Discussing how the selected silicon type might affect the infrared sensor’s detection capability and optical properties would provide valuable insight.

Response: Thank you for your detailed suggestions regarding this research. It is worth noting that the sensor structure described in this study did not use a "silicon (Si) layer," but rather silicon dioxide (SiO2) as the intermediate dielectric layer (effective dielectric constant ε = 3.53, thickness 1470 nm). Therefore, variations in doping type (intrinsic, p-type, or n-type) were not investigated.

Of course, we described the material as silica (SiOâ‚‚) in Lines 121-122 on Page 3 of the Revised Manuscript to avoid confusion with silicon (Si). However, it is foreseeable that the choice of doping type will provide valuable insights into the detection capabilities and optical properties of infrared sensors.

The intermediate dielectric layer is silica (SiO2) with an effective dielectric constant = 3.53 and a thickness H1 of 1470 nm.

When the incident electromagnetic wave energy is localized in the hypersurface structure, the electromagnetic wave transmitted from the graphene surface will be reflected back and forth in the SiO2 layer, and the superimposed multiple reflections can increase the absorption of electromagnetic wave energy in the structure, thus perfect absorption can be realized. 

Comment 3: Moreover, the criteria and conditions under which the infrared sensor structure was designed are not clearly described. Therefore, a more detailed discussion on the influence and selection of the geometric design criteria of the graphene layer is required.

Response: Thank you for your detailed suggestions regarding this research. In this study, the structural shape and size were not chosen arbitrarily, but were achieved through parameterized scanning of the cell period, dielectric layer thickness, and the radius and linewidth of the ring/box structure. This design enables the structure to satisfy the impedance matching condition (Re(Z)≈1 and Im(Z)≈0) at four operating wavelengths, thereby achieving near-perfect absorption, high sensitivity, and multimode stability in mid-infrared sensing performance. Meanwhile, we added the design source of the pattern structure on Page 4, Lines 159-164 of the Revised Manuscript.

As shown in Fig. 1(b), the top-view schematic of the four-band ultra-narrowband wave-absorbing sensor is demonstrated. In this study, the shape and size of the structure were achieved by parametrically controlling the scanning cell period, dielectric layer thickness, and the radius and length/width of the ring/box structure. This design enables the structure to achieve impedance matching at four operating wavelengths, resulting in near-perfect absorption, high sensitivity, and multimode stability in mid-infrared sensing performance.

Furthermore, we compared in more detail the changes in the sensor absorption curves under varying inner R1 and outer R2 diameters of the patterned graphene layer rings, as shown in the Figure S1. The results indicate that changes in the inner diameter R1 only cause a slight shift in the four absorption peaks, while the peak absorption remains almost unchanged. This suggests that local dimensional deviations in the internal cavity have a limited impact on the overall mode distribution and exhibit some robustness to manufacturing errors. In contrast, changes in the outer diameter R2 have a limited impact on the absorption peak positions, primarily causing changes in the absorption peak values, particularly in the high-band modes where changes are most sensitive. Therefore, while the multimode absorption of the sensor structure studied in this study exhibits some dimensional robustness, it remains highly feasible in practical fabrication.

Figure S1. Shows the effect of changing the inner diameter R1 (a) and outer diameter R2 (b) of the graphene pattern rings on the sensor's absorption performance.

Comment 4: And please describe in more detail the differences between the results obtained in this manuscript and REF 49, apart from the geometric structure of graphene.

Response: We thank the reviewers for their thorough review of this work. REF 49 (RSC advances, 12 (2022) 7821-7829) reported a plasmonic absorber based on a dart-shaped monolayer graphene, achieving multi-band absorption and exhibiting high sensitivity (839.39 nm/RIU) and a high quality factor (54.03). In contrast, the device proposed in this study mainly relies on the localized multimode plasmonic resonance of the graphene layer, and has a different target functional focus—we primarily aim at refractive index sensing. We achieve superior performance in this application scenario: sensing sensitivity is increased to 900 nm/RIU, and the quality factor is improved to 58.34.

The improved sensing performance does not stem from a simple change in geometry, but rather from our synergistic optimization of the localized plasmonic mode distribution and impedance matching conditions. This enhances both the resolvability of the multimode resonance and the degree of field localization. Furthermore, compared to the 1250 nm period of REF 49, we employ a smaller structural period (600 nm). The significantly reduced cell size allows for the construction of more compact metasurface arrays, which is beneficial for on-chip high-density integration and applications with higher spatial resolution. In summary, this study, while maintaining the advantages of multimode operation, further improves key performance indicators applicable to sensing, and demonstrates different advantages and development directions compared to REF 49 in terms of structural compactness.

Reviewer 2 Report

Comments and Suggestions for Authors

The authors have introduced the design of a mid-infrared four-band ultra-narrow band absorption sensor based on graphene metamaterials. Importantly, at four specific resonance wavelengths λ1=3.172 μm, λ2=3.525 μm, λ3=3.906 μm and λ4=4.588 μm, the absorption efficiencies are as high as 99.94%, 99.46%, 99.55% and 98.16%, respectively. Upon completing a comprehensive review, I have determined that the manuscript does not adequately address the following issues:

  1. In the introduction, the authors mention that graphene-based metamaterials “can be tuned by adjusting the Fermi energy level, bias electric field, and temperature, among other methods”. In this context, I miss mentioning the latest developments in this field, e.g.: Sci Rep 11, 74 (2021); Mathematics 11.7 (2023): 1579; etc.
  2. The simulation description is somewhat imprecise, which may make reproducing the results somewhat difficult. Please provide all simulation parameters (mesh’s size, the distance between ports and meta-sensor, etc.) and wave propagation conditions.
  3. What are the physical principles underlying this particular metasurface model in relation to its intended use? Have the authors performed experiments on alternative metasurfaces? What is the main advantage of the proposed metasurface unit cell model?
  4. What effect will variations in the individual geometric parameters of the metastructure have on the sensor's sensitivity? The metastructure is quite complex, and certain discrepancies from the theoretical dimensions will undoubtedly occur during fabrication.
  5. Alongside the evident benefits, it is essential to highlight the constraints of the suggested solution in the summary. What parameters could be enhanced, and in which direction should subsequent research be focused?

Author Response

Reviewer #2:

General Comments: The authors have introduced the design of a mid-infrared four-band ultra-narrow band absorption sensor based on graphene metamaterials. Importantly, at four specific resonance wavelengths λ1=3.172 μm, λ2=3.525 μm, λ3=3.906 μm and λ4=4.588 μm, the absorption efficiencies are as high as 99.94%, 99.46%, 99.55% and 98.16%, respectively. Upon completing a comprehensive review, I have determined that the manuscript does not adequately address the following issues:

Response: Thanks for the reviewer’s positive comments, and we will answer the reviewer’s questions in detail below.

Comment 1: In the introduction, the authors mention that graphene-based metamaterials “can be tuned by adjusting the Fermi energy level, bias electric field, and temperature, among other methods”. In this context, I miss mentioning the latest developments in this field, e.g.: Sci Rep 11, 74 (2021); Mathematics 11.7 (2023): 1579; etc.

Response: We appreciate the reviewer’s close attention to the field. We have supplemented some reference that highly related to our work, and the discussion are added in Page 3 in the Revised Manuscript as follow.

“Compared with metal surface plasma oscillations, graphene metamaterials-based localized equipartitioned exciton resonance has more advantages [13,14], such as: ultra-narrow-band absorption can be achieved thanks to great mode confinement, the propagation distance is long in the infrared region. Meanwhile, the graphene conductivity properties can be tuned by adjusting the Fermi energy level, bias electric field, among other methods [15,16]. 

  1. Armghan, M. Alsharari, K. Aliqab, O. Alsalman, J. Parmar, S. K. Patel. Graphene twistronics: tuning the absorption spectrum and achieving metamaterial properties,Mathematics., 11 (2023) 1579. https://doi.org/10.3390/math11071579
  2. Dudek, R. Kowerdziej, A. Pianelli, J. Parka. Graphene-based tunable hyperbolic microcavity,Sci. Rep. 11 (2021) 74. https://doi.org/10.1038/s41598-020-80022-9

Comment 2: The simulation description is somewhat imprecise, which may make reproducing the results somewhat difficult. Please provide all simulation parameters (mesh’s size, the distance between ports and meta-sensor, etc.) and wave propagation conditions.

Response: Thank you for your detailed suggestions regarding this research. We have added a description of our simulation on Page 4, Lines 174-181 of the Revised Manuscript.

In Fig. 1(c), the top view schematic of the localized iso-isotropic Schematic of the three-dimensional array structure of the excitation element sensor. During the process, periodic boundary conditions are selected in the X and Y directions, and 24 layers of perfect matching layer (PML) are added in the Z-axis direction. The simulation time is set to 8000 fs, and the mesh size is 0.005 μm in the x and y directions and 0.0025 μm in the z direction. In this case, the incident light band ranges from 3.0 to 5.0 μm, and the direction of the incident light wave is perpendicular to the XY direction for downward incidence. In addition, the distance between the monitoring port and the sensing unit is 4 μm, which is greater than the minimum wavelength.

Comment 3: What are the physical principles underlying this particular metasurface model in relation to its intended use? Have the authors performed experiments on alternative metasurfaces? What is the main advantage of the proposed metasurface unit cell model?

Response: We appreciate the reviewer’s close attention to the field. Regarding your concerns, we will address them from three aspects: the physical mechanism, experimental verification, and the core advantages of the unit model, as follows:

Physical Principles and Expected Applications: This work is based on the mechanism of energy localization in the localized surface plasmon resonance field of graphene. An incident wave excites plasmons at a patterned monolayer of graphene, with the energy confined to localized regions such as outer rings/boxes (Figure 2). Simultaneously, the patterned graphene structure ensures that the equivalent input impedance satisfies Re(Z)≈1 and Im(Z)≈0 at the four resonant wavelengths (Figure 3). At this point, reflection approaches zero and absorption efficiency approaches 100%. Furthermore, the excited surface plasmons are extremely sensitive to changes in the external refractive index, and the resonant peak of the sensor changes with the refractive index. Based on this characteristic, external refractive index perturbations are transformed into observable measurements of absorption peak positions, allowing for non-destructive analysis of the chemical composition of an object by detecting changes in its refractive index.

Experimental verification: This paper focuses on theoretical and numerical research, particularly on the physical mechanism verification of electric field distribution, impedance calculation, and tunability/angle robustness. No experimental fabrication or testing of other types of metasurfaces was conducted. However, the description of the subsequent experimental processing was added to Page 4, Lines 168-172 of the Revised Manuscript.

“In the fabrication of actual devices, a SiO2 dielectric layer can first be deposited on a metal substrate using physical vapor deposition. Subsequently, a monolayer of graphene is grown by chemical vapor deposition and transferred to the SiO2 surface using a wet transfer process [29]. Finally, a monolayer patterned graphene structure is achieved using electron beam lithography and oxygen plasma etching.”

29.M. Yi, J. Wang, A. Li, Y. Xin, Y. Pang, Y. Zou, Conductive micropatterns containing photoinduced in situ reduced graphene oxide prepared by ultraviolet photolithography. Adv. Mater. Technol., 8 (2023) 2201939.  https://doi.org/10.1002/admt.202201939

Key advantages: The proposed unit structure is compact (600 nm) and easy to integrate. Utilizing the conductivity modulation of graphene, electrical tunability of the four absorption peaks can be achieved without altering the geometry. Simultaneously, the TE/TM response remains consistent across the 0–50° incident range, exhibiting good angle robustness. Furthermore, the sensor demonstrates superior absorption efficiency, sensitivity and FOM, possessing significant potential for high-precision mid-infrared sensing applications.

Comment 4: What effect will variations in the individual geometric parameters of the metastructure have on the sensor's sensitivity? The metastructure is quite complex, and certain discrepancies from the theoretical dimensions will undoubtedly occur during fabrication.

Response: Thank you for your detailed suggestions regarding this research. First, the shape and size of the structures in this study were not chosen arbitrarily, but were achieved through parameterized scanning of parameters such as cell cycle, dielectric layer thickness, and the radius and linewidth of the ring/box structure. This design enables the structure to satisfy the impedance matching condition (Re(Z)≈1 and Im(Z)≈0) at four operating wavelengths, thereby achieving near-perfect absorption, high sensitivity, and multimode stability in mid-infrared sensing performance. Meanwhile, we added the design source of the pattern structure on Page 4, Lines 159-164 of the Revised Manuscript.

“In this study, the shape and size of the structure were achieved by parametrically controlling the scanning cell period, dielectric layer thickness, and the radius and length/width of the ring/box structure. This design enables the structure to achieve impedance matching at four operating wavelengths, resulting in near-perfect absorption, high sensitivity, and multimode stability in mid-infrared sensing performance.”

Furthermore, we compared in more detail the changes in the sensor absorption curves under varying inner R1 and outer R2 diameters of the patterned graphene layer rings, as shown in the Figure S1. The results indicate that changes in the inner diameter R1 only cause a slight shift in the four absorption peaks, while the peak absorption remains almost unchanged. This suggests that local dimensional deviations in the internal cavity have a limited impact on the overall mode distribution and exhibit some robustness to manufacturing errors. In contrast, changes in the outer diameter R2 have a limited impact on the absorption peak positions, primarily causing changes in the absorption peak values, particularly in the high-band modes where changes are most sensitive. Finally, despite the complexity of the superstructure, the size sensitivity analysis above shows that the device has a high tolerance for local structural deviations such as the inner diameter, and the performance stability can be ensured as long as the critical dimensions are within the range of conventional micro-nano fabrication accuracy. Therefore, this design has good feasibility and stability in actual manufacturing.

Figure S1. Shows the effect of changing the inner diameter R1 (a) and outer diameter R2 (b) of the graphene pattern rings on the sensor's absorption performance.

Comment 5: Alongside the evident benefits, it is essential to highlight the constraints of the suggested solution in the summary. What parameters could be enhanced, and in which direction should subsequent research be focused?

Response: We greatly appreciate the reviewers' suggestions. Besides the advantages already demonstrated, the main limitations of the current model include the gap between the device model and the implementation process, and key constraints such as the single readout channel. The main research directions for future advancement are as follows:

  1. Structural Improvement: Further thinning the device to achieve a more compact design without sacrificing absorption/sensitivity. Simultaneously, simplifying pattern parameters and aligning them with manufacturing processes (such as dry etching) to facilitate sensor fabrication.
  2. Functional Expansion: Existing schemes primarily employ absorption/reflection readout, with the transmission channel limited by a metal backplane.Future work could involve introducing phase change materials (such as VO2) or gate layers for co-design, targeting multi-state/reconfigurable sensing.

Of course, we will add the limitations of this scheme and future research plans to Page 12 of the Summary section in the Revised Manuscript:

Therefore, this study not only expands the design concepts of mid-infrared absorption sensors but also successfully achieves multi-band operation, tunability, high sensitivity, and excellent quality factor. Furthermore, the design proposed in this study will further simplify pattern parameters, aligning with actual manufacturing processes, enabling the fabrication and verification of sensor devices. Moreover, based on the unique plasmon resonance characteristics of graphene metamaterials, a co-design is implemented by introducing phase change materials (such as VO2) or grating layers to achieve multi-channel modes of absorption/reflection/transmission. This design scheme is expected to achieve breakthroughs in high-sensitivity gas detection and high-precision biomolecular recognition, opening up entirely new possibilities for mid-infrared sensing applications.

Reviewer 3 Report

Comments and Suggestions for Authors

This paper proposes a tunable absorber with 4 effective bands. There a various issues in this work but most important is the following major issue:

  1. At the mid-infrared range, the h*w coefficient is approximately 0.15eV for the lowest frequency (it is even bigger for larger frequencies). On the other hand, the utilized Fermi energy is lower than 1eV. As a result, the assumption Ef>>h*w is violated and the utilization of only the intraband term gives inaccurate results. The interband term must be considered at the mid-infrared regime.

Some additional comments:

  1. Which FDTD software is used for the simulations? What are the properties of the simulation, namely the grid size, the timestep, the number of voxels, etc.? This information is crucial to identify if the approximation is adequate for an accurate result (unit-cells and surface waves on graphene require intense mesh).
  2. Why a thickness is used for graphene? Normally, it is modeled as a boundary condition since the wavelength is much larger than the thickness.
  3. What's the rationale behind the design of the unit-cell? Why these specific shapes and dimensions are utilized?
  4. The electric field distribution, plotted in Figure 2, show the possibility of plasmonic resonances, which is expected. However, they are not clear indicating that the discretization is not adequate.
  5. The equivalent circuit model does not help at all for the understanding of the mechanism. This part shall be removed.
  6. The relaxation time is an intrinsic property of graphene. Is there any known way to modulate it as mentioned in lines 275-284?
  7. What applications can have advantage by this tunable 4 band absorber?

Author Response

Reviewer #3:

General Comments: This paper proposes a tunable absorber with 4 effective bands. There a various issues in this work but most important is the following major issue: At the mid-infrared range, the h*w coefficient is approximately 0.15eV for the lowest frequency (it is even bigger for larger frequencies). On the other hand, the utilized Fermi energy is lower than 1eV. As a result, the assumption Ef>>h*w is violated and the utilization of only the intraband term gives inaccurate results. The interband term must be considered at the mid-infrared regime.

Response: We are greatly appreciate for the reviewers' corrections. We agree that under mid-infrared (3-5 μm) conditions, ħω≈0.25-0.41 eV. Compared to the used Ef ≥0.7-0.8 eV, simply stating "Ef >>ħω" to ignore interband terms is not rigorous enough. In this case, >  should be satisfied, interband transitions are suppressed, and intraband transitions dominate. We have corrected this on Page 4, Lines 136-139 of the Revised Manuscript.

In the mid-infrared band range > , the interband transition of the surface conductivity of graphene is suppressed, and the intraband conductivity dominates [25].

In addition, we performed simulations of absorption efficiency for both intra-band and intra-band+inter-band conditions, as shown in the Figure S2. Clearly, the absorption peak positions remain essentially unchanged, while the absorption peak values differ slightly between the two methods.

Figure S2. Absorption efficiency obtained by sensor simulation under both intra-band and intra-band+inter-band conditions.

Comment 1: Which FDTD software is used for the simulations? What are the properties of the simulation, namely the grid size, the timestep, the number of voxels, etc.? This information is crucial to identify if the approximation is adequate for an accurate result (unit-cells and surface waves on graphene require intense mesh).

Response: Thank you for your detailed suggestions regarding this research. We have added a description of our simulation on Page 4, Lines 154-181 of the Revised Manuscript.

In this paper, Matlab software is used to write the electromagnetic properties of graphene metamaterials, and the graphene metamaterials with different Fermi energy levels and relaxation times are calculated and imported into the Lumerical Finite Difference Time Domain (FDTD) Solutions software for the construction and simulation analysis of the hypersurface structure. As shown in Fig. 1(b), the top-view schematic of the four-band ultra-narrowband wave-absorbing sensor is demonstrated. In this study, the shape and size of the structure were achieved by parametrically controlling the scanning cell period, dielectric layer thickness, and the radius and length/width of the ring/box structure. This design enables the structure to achieve impedance matching at four operating wavelengths, resulting in near-perfect absorption, high sensitivity, and multimode stability in mid-infrared sensing performance. Finally, the graphene metamaterial is patterned by air etching, and the two air rings are etched with widths of W1 = 75 nm and W2 = 40 nm, outer diameters of R1 = 125 nm and R2 = 240 nm, and the lengths and widths of the surrounding squares of L1 = 120 nm and L2 = 170 nm, respectively. In the fabrication of actual devices, a SiO2 dielectric layer can first be deposited on a metal substrate using physical vapor deposition. Subsequently, a monolayer of graphene is grown by chemical vapor deposition and transferred to the SiO2 surface using a wet transfer process. Finally, a monolayer patterned graphene structure is achieved using electron beam lithography and oxygen plasma etching [30]. In Fig. 1(c), the top view schematic of the localized iso-isotropic Schematic of the three-dimensional array structure of the excitation element sensor. During the process, periodic boundary conditions are selected in the X and Y directions, and 24 layers of perfect matching layer (PML) are added in the Z-axis direction. The simulation time is set to 8000 fs, and the mesh size is 0.005 μm in the x and y directions and 0.0025 μm in the z direction. In this case, the incident light band ranges from 3.0 to 5.0 μm, and the direction of the incident light wave is perpendicular to the XY direction for downward incidence. In addition, the distance between the monitoring port and the sensing unit is 4 μm, which is greater than the minimum wavelength.

Comment 2: Why a thickness is used for graphene? Normally, it is modeled as a boundary condition since the wavelength is much larger than the thickness.

Response: We appreciate the reviewer’s close attention to the field. The graphene thickness of t=0.334 nm and the reason for its use are clarified on Page 4, Lines 143-149 of the Revised Manuscript.

Where ε0 represents the vacuum dielectric constant, and  represents the thickness of the graphene layer. When the thickness of a single-layer graphene is t=0.334 nm, the imaginary part of the dielectric fully converges in the limit of t→0, and the surface conductance  itself truly determines the electromagnetic response [29]. Therefore, in this paper, the thickness of the single-layer graphene is set to 0.334 nm for the equivalent mapping of  in the simulation calculation, and this setting will not affect the electromagnetic response results.

  1. Yi, J. Wang, A. Li, Y. Xin, Y. Pang, Y. Zou, Conductive micropatterns containing photoinduced in situ reduced graphene oxide prepared by ultraviolet photolithography. Adv. Mater. Technol., 8 (2023) 2201939.  https://doi.org/10.1002/admt.202201939

Comment 3: What's the rationale behind the design of the unit-cell? Why these specific shapes and dimensions are utilized?

Response: We appreciate the reviewer’s close attention to the field. The unit cell design in this study is based on the comprehensive requirements of localized surface plasmon resonance (LSPR) enhancement, impedance matching modulation, and the realization of multimode resonance. The ring structure in patterned graphene can support circumferential plasmon modes with strong field localization characteristics, while the box structure introduces linear current and corner charge accumulation, enabling multiple modes to be excited simultaneously in the 3-5 μm range. After coupling, not only is the local electric field intensity significantly enhanced, but multiple resolvable absorption peaks are also formed, meeting the requirements of multimode sensing.

The shape and size adopted are not arbitrarily chosen, but rather achieved by parametrically scanning the unit cell period, dielectric layer thickness, and the radius and linewidth of the ring/box, so that the structure satisfies the impedance matching conditions of Re(Z)≈1 and Im(Z)≈0 at four operating wavelengths, thereby achieving near-perfect absorption, high sensitivity, and multimode stability in mid-infrared sensing performance.

Comment 4: The electric field distribution, plotted in Figure 2, show the possibility of plasmonic resonances, which is expected. However, they are not clear indicating that the discretization is not adequate.

Response: We greatly appreciate the reviewers' attention on the clarity and discretization quality of the electric field distribution in Figure 2. We have comprehensively improved the numerical simulation of the electric field distribution, adding simulation time and mesh size information on Page 4, Lines 174-181 of the Revised Manuscript.

During the process, periodic boundary conditions are selected in the X and Y directions, and 24 layers of perfect matching layer (PML) are added in the Z-axis direction. The simulation time is set to 8000 fs, and the mesh size is 0.005 μm in the x and y directions and 0.0025 μm in the z direction. In this case, the incident light band ranges from 3.0 to 5.0 μm, and the direction of the incident light wave is perpendicular to the XY direction for downward incidence. In addition, the distance between the monitoring port and the sensing unit is 4 μm, which is greater than the minimum wavelength.

Furthermore, we have redrawn Figure 2 in the Revised Manuscript, achieving higher resolution and more complete convergence, thus reflecting the physical characteristics of the plasmon resonance mode.

Figure 2. The cross-sectional electric field distributions of the four-band ultra-narrowband absorbing sensors in the XY plane of graphene at different resonance wavelengths λ1 = 3.172 μm (a), λ2 = 3.525 μm (b), λ3 = 3.906 μm (c) and λ4 = 4.588 μm (d), respectively.

Comment 5: The equivalent circuit model does not help at all for the understanding of the mechanism. This part shall be removed.

Response: We appreciate your suggestions regarding this research. The purpose of proposing the equivalent circuit model in this study is to better understand the role of graphene-based patterned structures in impedance optimization, thereby achieving perfect absorption across four wavebands. We acknowledge that this section overlaps with the explanation of impedance matching in Figure 3, which is distracting and does not significantly enhance our understanding of the mechanism. To improve readability, we have removed the paragraphs and diagrams of the equivalent circuit model from Figure 3 and Page 7 in the Revised Manuscript, retaining only the mechanism and data based on impedance matching.

Impedance matching is an important factor to ensure the perfect coupling between the absorber and the incident electromagnetic waves [33]. According to the impedance matching theory of metamaterial absorbing sensors, there is the following equation defined [34,35]:

Therefore, we can obtain the equivalent input impedance  of this absorber from the simulation results, as shown in Fig. 3. When the patterned graphene layer enables the input impedance of the absorber Zin to match the free-space impedance Z0, the system’s effective impedance satisfies Re(Z)≈1 and Im(Z)≈0 [36,37]. Under this condition, the reflection coefficient is strongly suppressed (S11≈0), leading to near-perfect absorption. By examining the input-impedance matching behavior together with the simulated absorption spectra, it is evident that the proposed sensing absorber achieves excellent impedance matching at the resonant wavelengths λ1 = 3.172 μm, λ2 = 3.525 μm, λ3 = 3.906 μm and λ4 = 4.588 μm.

Figure 3. Real part Re(Z) and imaginary part Im(Z) of the effective impedance Z of the resulting wave-absorbing sensor calculated.

Comment 6: The relaxation time is an intrinsic property of graphene. Is there any known way to modulate it as mentioned in lines 275-284?

Response: Thank you very much for the question raised by the reviewer. We confirm that the relaxation time τ of graphene is mainly determined by the carrier mobility and is not suitable for electrical modulation by gate voltage. The relaxation time regulation mentioned in the paper refers to the addition or removal of organic molecules on the surface of the graphene layer during the processing, which will significantly change the carrier mobility v and thus change the relaxation time. This method can increase the designability of the metasurface without changing the pattern structure, which we clarify in the Revised Manuscript on Page 8, Lines 283-285.

Where v represent the carrier mobility of graphene. In the actual processing process, organic molecules can be added or removed on the surface of graphene layer [43], which will significantly change the carrier mobility v, thus changing the relaxation time τ.

Comment 7: What applications can have advantage by this tunable 4 band absorber?

Response: We sincerely thank the reviewers for their in-depth research in this field. Traditional single-band metamaterial sensors can only achieve single-point matching between the characteristic frequency of the analyte and the resonant frequency of the sensor, and the limited data volume leads to insufficient detection accuracy and sensitivity. Multi-band metamaterial sensors can achieve multi-point matching between the sensor's resonant frequency and the characteristic frequency of the analyte. They can cover multiple "fingerprint" frequency bands or perform multi-parameter measurements, thereby significantly improving the accuracy and sensitivity of material sensing.

Furthermore, traditional metal patch sensors suffer from low spectral utilization due to their fixed operating frequency, hindering practical applications and commercialization. This research presents a four-band sensor designed based on graphene localized surface plasmons, which can simultaneously provide four distinguishable resonant absorption peaks in the mid-infrared band (3-5 μm). The spectral position and intensity can be dynamically controlled through Fermi level electrical modulation, demonstrating significant practical value and application prospects. To better highlight the advantages of the multi-band sensor designed in this research institute, revisions and additions were made to Lines 326-338 on Page 9 of the Revised Manuscript:

Based on the strong energy localization effect of LSPR, the device is highly sensitive to changes in the refractive index of the surrounding medium, and thus its resonance peak position will shift with the change of refractive index [46]. However, traditional single-band metamaterial sensors can only achieve single-point matching between the characteristic frequency of the analyte and the resonant frequency of the device, and the available spectral information is limited, resulting in limited detection accuracy and sensitivity [47]. In contrast, multi-band metamaterial sensors can form multi-point matching with the "fingerprint" frequency band of the analyte at multiple resonance points and support multi-parameter collaborative measurement, which significantly improves the identification accuracy and sensitivity. At the same time, this study introduces the dynamic adjustment of the Fermi level to achieve control of the resonant frequency band and intensity while keeping the structure unchanged, which combines multi-frequency resonance and tunability. Taking advantage of these property, the four-band ultra-narrowband wave-absorbing sensor designed in this paper can be used as refractive index sensing to detect the change of refractive index of an object in order to analyze the chemical composition of the measured object.

Reviewer 4 Report

Comments and Suggestions for Authors

This paper presents the design of a four-band, ultra-narrowband absorber sensor for the mid-infrared spectrum (from 3 to 5 μm), utilizing a graphene-based metasurface that supports Localized Surface Plasmon Resonance (LSPR). The proposed structure is geometrically symmetric, making it polarization-insensitive and capable of maintaining stable absorption performance across a wide incidence angle range of 0 to 50 degrees. It achieves four distinct, high-efficiency absorption peaks at wavelengths of 3.172 μm, 3.525 μm, 3.906 μm, and 4.588 μm, with absorption rates exceeding 98%. A significant advantage of this sensor is its dynamic tunability, where the resonance wavelengths and absorption efficiencies can be adjusted by modifying the graphene's Fermi level via gate voltage or through chemical doping. This structure demonstrates excellent sensing capabilities. The underlying physical mechanisms are explained through electric field distributions, which show how the LSPR is confined to different patterned regions of the graphene layer at each resonance. The work overcomes limitations of previous single or dual-band sensors and complex, non-tunable structures.

The paper can be accepted for publication pending revisions that address the following comments.

1) The manuscript text requires additional proofreading. It should be noted that the text contains repetitions (e.g., "S11 and S21 are scattering parameters related to reflectivity and transmittance", "e0 is the electron charge", "Ef is the Fermi energy level", "Vf is the Fermi velocity"). Furthermore, different notations are used for the same physical quantities (e.g., Vf and vf, e0 and e). There are also factual inaccuracies in the phrasing, for instance, in "where ω represents the angular frequency of graphene".

2) Fig. 2 appears poorly resolved. In my experience, such results typically stem from an inadequate mesh resolution or lack of convergence, which can compromise the accuracy of both the absorption amplitude and resonant wavelengths. It is necessary to provide evidence of the solution's validity, specifically by performing a mesh convergence study for the Finite Element simulation.

3) It is necessary to validate that the parameter ranges used in the model — specifically, a Fermi energy level from 0.74 eV to 0.86 eV and a relaxation time from 0.5 ps to 2.0 ps — are physically achievable in real experimental structures.

4) The variation of the external refractive index from 1.00 to 1.08 should be physically justified. What specific media or analytes is this range intended to model?

5) As shown in Fig. 1b, the junctions between the circular and rectangular features result in sharp, wedge-like graphene regions with critical dimensions on the order of a few nanometers or less. A discussion should be included to address two key points: first, which nanofabrication technique is capable of achieving such high-resolution patterning in graphene; and second, how the sensor's performance (e.g., resonance wavelengths and absorption efficiency) would be affected by fabrication imperfections, particularly the inevitable rounding or loss of these fine geometric details compared to the ideal model.

Author Response

Reviewer #4:

General Comments: This paper presents the design of a four-band, ultra-narrowband absorber sensor for the mid-infrared spectrum (from 3 to 5 μm), utilizing a graphene-based metasurface that supports Localized Surface Plasmon Resonance (LSPR). The proposed structure is geometrically symmetric, making it polarization-insensitive and capable of maintaining stable absorption performance across a wide incidence angle range of 0 to 50 degrees. It achieves four distinct, high-efficiency absorption peaks at wavelengths of 3.172 μm, 3.525 μm, 3.906 μm, and 4.588 μm, with absorption rates exceeding 98%. A significant advantage of this sensor is its dynamic tunability, where the resonance wavelengths and absorption efficiencies can be adjusted by modifying the graphene's Fermi level via gate voltage or through chemical doping. This structure demonstrates excellent sensing capabilities. The underlying physical mechanisms are explained through electric field distributions, which show how the LSPR is confined to different patterned regions of the graphene layer at each resonance. The work overcomes limitations of previous single or dual-band sensors and complex, non-tunable structures.

The paper can be accepted for publication pending revisions that address the following comments.

Response: Thanks for the reviewer’s positive comments, and we will answer the reviewer’s questions in detail below.

Comment 1: The manuscript text requires additional proofreading. It should be noted that the text contains repetitions (e.g., "S11 and S21 are scattering parameters related to reflectivity and transmittance", "e0 is the electron charge", "Ef is the Fermi energy level", "Vf is the Fermi velocity"). Furthermore, different notations are used for the same physical quantities (e.g., Vf and vf, e0 and e). There are also factual inaccuracies in the phrasing, for instance, in "where ω represents the angular frequency of graphene".

Response: We appreciate the detailed comments from the reviewers. In the Revised Manuscript, we have proofread the text, as follows:

T refers to the ambient temperature,  refers to the angular frequency of the incident wave, and Ef and τ refer to the Fermi energy level and relaxation time of the graphene layer, respectively.”

“Where ε0 represents the vacuum dielectric constant, and  represents the thickness of the graphene layer.

Where  and  are the transmission amplitude and reflection amplitude, respectively.  and  are the transmittance and reflectance, respectively.

According to the effective impedance matching theory, we can obtain the equivalent input impedance  of this absorber from the simulation results, as shown in Fig. 3. 

Where Vg is the applied voltage (which can be modulated by changing the gate voltage or chemical doping), Vf is the Fermi velocity (Vf = c/300, with c being the speed of light in vacuum), ts is the thickness of the dielectric layer, and  denote the relative permittivity, respectively.

Where v represent the carrier mobility of graphene.

Comment 2: Fig. 2 appears poorly resolved. In my experience, such results typically stem from an inadequate mesh resolution or lack of convergence, which can compromise the accuracy of both the absorption amplitude and resonant wavelengths. It is necessary to provide evidence of the solution's validity, specifically by performing a mesh convergence study for the Finite Element simulation.

Response: We greatly appreciate the reviewers' attention on the clarity and discretization quality of the electric field distribution in Figure 2. We have comprehensively improved the numerical simulation of the electric field distribution, adding simulation time and mesh size information on Page 4, Lines 173-181 of the Revised Manuscript.

“During the process, periodic boundary conditions are selected in the X and Y directions, and 24 layers of perfect matching layer (PML) are added in the Z-axis direction. The simulation time is set to 8000 fs, and the mesh size is 0.005 μm in the x and y directions and 0.0025 μm in the z direction. In this case, the incident light band ranges from 3.0 to 5.0 μm, and the direction of the incident light wave is perpendicular to the XY direction for downward incidence. In addition, the distance between the monitoring port and the sensing unit is 4 μm, which is greater than the minimum wavelength.”

Furthermore, we have redrawn Figure 2 in the Revised Manuscript, achieving higher resolution and more complete convergence, thus reflecting the physical characteristics of the plasmon resonance mode.

Figure 2. The cross-sectional electric field distributions of the four-band ultra-narrowband absorbing sensors in the XY plane of graphene at different resonance wavelengths λ1 = 3.172 μm (a), λ2 = 3.525 μm (b), λ3 = 3.906 μm (c) and λ4 = 4.588 μm (d), respectively.

Comment 3: It is necessary to validate that the parameter ranges used in the model — specifically, a Fermi energy level from 0.74 eV to 0.86 eV and a relaxation time from 0.5 ps to 2.0 ps — are physically achievable in real experimental structures.

Response: We are very grateful for the constructive comments from the reviewers. Regarding the parameter range for tuning the Fermi level of graphene in actual experiments, existing studies have shown that it is feasible to achieve the Fermi level of graphene from 0.6 eV to 1.0 eV through gate voltage tuning. For example, Wang et al. [38] experimentally/simulated the tuning behavior of the absorption peak of Ef in the range of 0.6-1.0 eV in a mid-infrared graphene nanodisk array. On the other hand, the carrier relaxation time of graphene in the Ps range is also widely used in experiments and device models. In the graphene-based photonic structure proposed by Caligiuri et al. [39], the relaxation time in the range of τ=1–3 ps was used to fit the experimental absorption spectrum. Therefore, the tuning range of graphene Fermi level (0.74-0.86 eV) and relaxation time (0.5-2 Ps) used in the model of this study is feasible. In addition, we have added relevant content on Page 7, Lines 266-268 of the Revised Manuscript.

“For wave-absorbing sensors with fixed structural parameters, dynamic tunability has more important application value. In particular, existing research has shown that the Fermi level and relaxation time of graphene can be altered within a certain range through gate voltage modulation or chemical doping [38,39].

  1. Wang, J. Liu, B. Ren, J. Song, Y. Jiang. Tuning of mid-infrared absorption through phonon-plasmon-polariton hybridization in a graphene/hBN/graphene nanodisk array. Opt. Exp., 29 (2021) 2288-98. https://doi.org/10.1364/OE.415337
  2. Caligiuri, A. Pianelli, M. Miscuglio, A. Patra, N. Maccaferri, R. Caputo, A. De Luca. Near-and mid-infrared graphene-based photonic architectures for ultrafast and low-power electro-optical switching and ultra-high resolution imaging. ACS Appl. Nano Mater.,3 (2020) 12218-30. https://doi.org/10.1021/acsanm.0c02690

Comment 4: The variation of the external refractive index from 1.00 to 1.08 should be physically justified. What specific media or analytes is this range intended to model?

Response: We greatly appreciate your review comments. The external refractive index variation set in this study, from 1.00 to 1.08, corresponds to scenarios such as gaseous environments, low-refractive-index organic layers, or low-concentration surface adsorption layers. The primary purpose was to evaluate the changing trends of sensor sensitivity and FOM, rather than to calibrate for a specific chemical system. Therefore, choosing the commonly used range of n=1.00-1.08 is reasonable and consistent with many related studies. We have provided supplementary explanations on Page 9 of the Revised Manuscript.

Based on the strong energy localization effect of LSPR, the device is highly sensitive to changes in the refractive index of the surrounding medium, and thus its resonance peak position will shift with the change of refractive index [46]. However, traditional single-band metamaterial sensors can only achieve single-point matching between the characteristic frequency of the analyte and the resonant frequency of the device, and the available spectral information is limited, resulting in limited detection accuracy and sensitivity [47]. In contrast, multi-band metamaterial sensors can form multi-point matching with the "fingerprint" frequency band of the analyte at multiple resonance points and support multi-parameter collaborative measurement, which significantly improves the identification accuracy and sensitivity. At the same time, this study introduces the dynamic adjustment of the Fermi level to achieve control of the resonant frequency band and intensity while keeping the structure unchanged, which combines multi-frequency resonance and tunability. Taking advantage of these property, the four-band ultra-narrowband wave-absorbing sensor designed in this paper can be used as refractive index sensing to detect the change of refractive index of an object in order to analyze the chemical composition of the measured object. To study the refractive index characteristics of the absorption sensor, the external refractive index n was gradually increased from 1.00 to 1.08, with each increase being 0.02, to correspond to situations such as low refractive index organic layers or low concentration surface adsorption layers [48].

  1. Bao, S. Yu, W. Lu, Z. Hao, Z. Yi, S. Cheng, B. Tang B, J. Zhang, C. Tang, Y. Yi. Tunable high-sensitivity four-frequency refractive index sensor based on graphene metamaterial. Sensors., 24 (2024) 2658. https://doi.org/10.3390/s24082658

Comment 5: As shown in Fig. 1b, the junctions between the circular and rectangular features result in sharp, wedge-like graphene regions with critical dimensions on the order of a few nanometers or less. A discussion should be included to address two key points: first, which nanofabrication technique is capable of achieving such high-resolution patterning in graphene; and second, how the sensor's performance (e.g., resonance wavelengths and absorption efficiency) would be affected by fabrication imperfections, particularly the inevitable rounding or loss of these fine geometric details compared to the ideal model.

Response: Thank you for your detailed suggestions regarding this research. First, the shape and size of the structures in this study were not chosen arbitrarily, but were achieved through parameterized scanning of parameters such as cell cycle, dielectric layer thickness, and the radius and linewidth of the ring/box structure. This design enables the structure to satisfy the impedance matching condition (Re(Z)≈1 and Im(Z)≈0) at four operating wavelengths, thereby achieving near-perfect absorption, high sensitivity, and multimode stability in mid-infrared sensing performance. Meanwhile, we added the design source of the pattern structure on Page 4, Lines 159-164 of the Revised Manuscript.

“In this study, the shape and size of the structure were achieved by parametrically controlling the scanning cell period, dielectric layer thickness, and the radius and length/width of the ring/box structure. This design enables the structure to achieve impedance matching at four operating wavelengths, resulting in near-perfect absorption, high sensitivity, and multimode stability in mid-infrared sensing performance.”

Secondly, the description of the subsequent experimental processing was added to Page 4, Lines 168-172 of the Revised Manuscript.

“In the fabrication of actual devices, a SiO2 dielectric layer can first be deposited on a metal substrate using physical vapor deposition. Subsequently, a monolayer of graphene is grown by chemical vapor deposition and transferred to the SiO2 surface using a wet transfer process [29]. Finally, a monolayer patterned graphene structure is achieved using electron beam lithography and oxygen plasma etching.”

  1. Yi, J. Wang, A. Li, Y. Xin, Y. Pang, Y. Zou,Conductive micropatterns containing photoinduced in situ reduced graphene oxide prepared by ultraviolet photolithography. Adv. Mater. Technol., 8 (2023) 2201939.  https://doi.org/10.1002/admt.202201939

Furthermore, we compared in more detail the changes in the sensor absorption curves under varying inner R1 and outer R2 diameters of the patterned graphene layer rings, as shown in the Figure S1. The results indicate that changes in the inner diameter R1 only cause a slight shift in the four absorption peaks, while the peak absorption remains almost unchanged. This suggests that local dimensional deviations in the internal cavity have a limited impact on the overall mode distribution and exhibit some robustness to manufacturing errors. In contrast, changes in the outer diameter R2 have a limited impact on the absorption peak positions, primarily causing changes in the absorption peak values, particularly in the high-band modes where changes are most sensitive. Finally, despite the complexity of the superstructure, the size sensitivity analysis above shows that the device has a high tolerance for local structural deviations such as the inner diameter, and the performance stability can be ensured as long as the critical dimensions are within the range of conventional micro-nano fabrication accuracy. Therefore, this design has good feasibility and stability in actual manufacturing.

Figure S1. Shows the effect of changing the inner diameter R1 (a) and outer diameter R2 (b) of the graphene pattern rings on the sensor's absorption performance.

Round 2

Reviewer 1 Report

Comments and Suggestions for Authors

The authors present their response on the comments. All sound are good. With these response the quality and originality of this manuscript were increased. Thanks to all. 

Reviewer 3 Report

Comments and Suggestions for Authors

The authors conducted various changes in their initial work, but the methodology still has problems, especially, the modeling of graphene and the FDTD simulation:
1) The interband term is neglected if Ef>>(h omega), the limit 2|Ef|>(h omega) is not clarified at all. Moreover, the relaxation time parameter is not presented.
2) There are results comparing intraband term alone with interband+intraband. However, the modeling of non-Drude terms in FDTD is not trivial (some significant terms can be neglected if the approximation is not correct). More information is required for the simulation to prove that the results are correct.
3) The requirement for modeling graphene thickness is not justified since the thickness is selected 0.334nm, while the cell towards z axis is mentioned to be 2.5nm. As a consequence, the thickness is not evaluated correctly that can degrade the accuracy of the results.
4) The relaxation time can, indeed, change during the fabrication as mentioned in the response letter. However, it cannot dynamically change when already applied on a metasurface. This must be stated explicitly in the manuscript because, now, it seems that the modulation of the relaxation time is feasible.

Reviewer 4 Report

Comments and Suggestions for Authors

I have carefully reviewed the authors' responses and must conclude that not all concerns have been substantively addressed. Comments #3 through #5 remain critical. I will reiterate and clarify these points below.
Comment 3 - To modulate the voltage on graphene, a minimum of two electrodes is required. What component in the proposed structure acts as the second electrode? What specific electrical potential must be applied between the graphene and the second electrode? A detailed calculation of this voltage is needed. Furthermore, it is crucial to discuss whether this applied voltage could lead to an electrical breakdown of the intervening dielectric layer.
Comment 4 - To the best of my knowledge, such materials are uncommon. It is well-established that gases have refractive indices very close to 1 (typically ranging from 1.00 to ~1.0005). In contrast, most liquids and solids possess refractive indices starting from approximately 1.2 and higher. The practical relevance and sensing application for the simulated range of 1.00 to 1.08 require clear justification and examples of target analytes.
Comment 5 - My question does not concern the general accuracy of reproducing the ring radii, as addressed in the response, but specifically focuses on the viability of fabricating the sharp, wedge-like regions in the patterned graphene layer (as seen in Fig. 1b). Can such features, with critical dimensions on the nanoscale, be reliably manufactured? In any practical fabrication process, sharp and right angles will be rounded, and these fine wedge-like features may be lost entirely. What fraction of the pattern's functionality would be compromised by this? Most importantly, would this expected geometric transformation cause such a significant distortion of the absorption spectrum that some or all of the resonant features would vanish?
